# Cocoa By-Products: Characterization of Bioactive Compounds and Beneficial Health Effects

**DOI:** 10.3390/molecules27051625

**Published:** 2022-03-01

**Authors:** Thiago F. Soares, M. Beatriz P. P. Oliveira

**Affiliations:** REQUIMTE/LAQV, Department of Chemical Sciences, Faculty of Pharmacy, University of Porto, R. J. Viterbo, 4050-313 Porto, Portugal; thiago.f.soares@outlook.com

**Keywords:** cocoa beans, cocoa by-products, circular economy, sustainability of the food industry, polyphenols, theobromine, methylxanthines

## Abstract

The annual production of cocoa is approximately 4.7 million tons of cocoa beans, of which only 10% corresponds to the cocoa bean and the remaining value corresponds to a high number of residues, cocoa bean shell, pulp and husk. These by-products are a source of nutrients and compounds of notable interest in the food industry as possible ingredients, or even additives. The assessment of such by-products is relevant to the circular economy at both environmental and economic levels. Investigations carried out with these by-products have shown that cocoa husk can be used for the production of useful chemicals such as ketones, carboxylic acids, aldehydes, furans, heterocyclic aromatics, alkylbenzenes, phenols and benzenediols, as well as being efficient for the removal of lead from acidic solutions, without decay in the process due to the other metals in this matrix. The fibre present in the cocoa bean shell has a considerable capacity to adsorb a large amount of oil and cholesterol, thus reducing its bioavailability during the digestion process, as well as preventing lipid oxidation in meats, with better results compared to synthetic antioxidants (butylated hydroxytoluene and β-tocopherol). Finally, cocoa pulp can be used to generate a sweet and sour juice with a natural flavour. Thus, this review aimed to compile information on these by-products, focusing mainly on their chemical and nutritional composition, simultaneously, the various uses proposed in the literature based on a bibliographic review of articles, books and theses published between 2000 and 2021, using databases such as Scopus, Web of Science, ScieLO, PubMed and ResearchGate.

## 1. Introduction

*Theobroma cacao* L., the cocoa-producing tree, belongs to the order of Malvales, family Malvaceae and genus *Theobroma*. Its name has Greek origins, Theos and Bromos, meaning “food of the gods” [1]. It is the only plant with commercial use in the production of chocolate. Despite having a pulp with a pleasant flavour, cocoa seed is the part of cocoa mostly used in the food industry, generating several products (Figure 1) with the greatest focus on chocolate [1].

There are some hypotheses for the origin and dispersion of cocoa, one of them that it comes from the upper region of the Orinoco and Amazon basins [3]. Cristovão Colombo was responsible for the knowledge and management of cocoa in the world since his arrival in America [3]. With the growing popularity of cocoa and its products, in 1824, the Portuguese transported some Forastero cocoa seedlings from Brazil to São Tomé e Príncipe. Around 1850, other seedlings went to Equatorial Guinea and, finally, in the mid-1900s, the seedlings spread to Côte d’Ivoire, Ghana and Nigeria, countries that became the world’s largest cocoa producers [3].

There are four main varieties of *Theobroma cacao* L.: Nacional, Criollo, Forastero and Trinitário [4]. Forastero is the most representative variety in world, with about 80% of production, as it is more resistant to diseases and has higher productivity [5]. Nacional is the rarest among the four varieties and is characterized by its more refined taste, being less bitter and more aromatic, thus having greater economic value [6].

The annual production of the cocoa industry is about 4.7 million tons of cocoa seeds [7], as shown in Figure 2A. According to data from the International Cocoa Organization (ICCO), Côte d’Ivoire was the largest cocoa producer in the 2018/2019 harvest with 2154 thousand tons, followed by Ghana (812 thousand tons) and Ecuador (322 thousand tons), as shown in Figure 2B. African countries are responsible for nearly 80% of the world’s cocoa seed production, although Nacional cocoa is found mainly in Ecuador and nearby regions.

## 2. Methodology

The used methodology to produce this review was based on a bibliographic review of articles, books and theses published between 2000 and 2021. However, some publications prior to 2000 were also considered due to their relevant contributions, or for being unique studies in the area of knowledge. The search engines and databases used were Scopus, Web of Science, ScieLO, PubMed and ResearchGate. The bibliographic sources were obtained by individual and collective searching of the terms: “cocoa bean shell”, “cocoa husk”, “cocoa pulp”, “by-product”, “polyphenols”, “theobromine”, “methylxanthines” and “dietary fibres”. It is also worth mentioning that if there were synonyms for cocoa by-products, they were used to improve the carried-out research.

## 3. Cocoa Transformation

The main activity of the cocoa industry is the transformation of cocoa fruits into cocoa mass. This process is carried out in two consecutive steps: pre-processing followed by processing [4]. The quality of the final product is closely related to these steps. An inadequate fermentation in the pre-processing step, for example, can negatively influence the chemical constituents and metabolites of the final product [4].

### 3.1. Cocoa Pre-Processing

After harvesting, the fruits are washed and opened manually, and the placenta separated from the seeds. This step allows a more adequate fermentation, generating seeds with better sensory characteristics. After removing the pulp, the seeds have about 65% moisture [8]. To avoid undesirable chemical reactions and the development of spoilage microorganisms, the interval between opening the fruit and fermentation should not exceed 24 h [9,10].

Fermentation lasts between 2 and 8 days, due to the type of cocoa and the environmental characteristics [1]. During this process, the seeds are stirred frequently to optimize air penetration and improve product homogeneity [11]. The pulp pH of around 3.6 and the low level of available oxygen create the ideal conditions for alcoholic fermentation in the presence of yeasts [8]. With penetration of oxygen in the fermentation process, over time, acetic bacteria convert the alcohol into acetic acid, reducing the pH and increasing the temperature (45–50 °C) [8]. The diffusion of acids in the seed and the increase in temperature kill the germ. Subsequently, seeds become cocoa beans and taste precursors, amino acids, peptides, and reducing sugars (the precursors of the Maillard reaction in the roasting phase) are also formed [12].

Finally, the cocoa beans go into drying and storage. There are two techniques for this step: natural or artificial. The natural technique is quite simple, requiring only the space available to perform it. The artificial one uses dryers in a warehouse [8]. One of the most important parameters to control is the water removal rate. If high, the inside of the cocoa beans will remain moist, promoting the proliferation of fungi/microorganisms and reducing the quality of the product. If slow, the product will lose weight, increasing the fragility of cocoa beans [13]. The drying stage is intended not only to eliminate the water in excess (final moisture close to 7%) [14], but also to reduce the acidity of the cocoa beans, following the biochemical changes initiated in the fermentation, which will give the characteristic flavour and aroma of chocolate [14]. This whole process takes place near the fruit harvest site.

### 3.2. Cocoa Processing

The processing stage already takes place at the company that purchases the raw material. The first step in this process is cleaning the dry cocoa beans, using sieves and magnets, to separate foreign materials (stones, metals and plant pieces). At the same time, shelling may occur [9]. The industry can use two different procedures in series: heat pre-treatment and the roasting of cocoa beans with subsequent shelling. The advantage of heat pre-treatment before roasting is the production of cocoa beans of similar sizes [9]. In this phase, equipment such as fluidized bed dryers, infrared, continuous air roasters, among others, can be used. Cocoa beans are usually broken due to impact at high speeds against obstacles [12]. After breaking, the product is transferred to vibrating sieves to separate the husks with the aid of a blow of air, due to the difference in weight [9]. The last by-product of cocoa, as shown in Figure 3, is obtained at this stage.

In the roasting process, the control of time and temperature (110–140 °C) is essential, as they are responsible for the characteristic flavour and taste of chocolate [11]. Roasting, as a heat treatment, reduces the water content and inactivates the enzymes in the product. Its main function is the development of the aroma, through Maillard reactions, initiated earlier in the fermentation step and responsible for the characteristic brown colour [10,14]. Finally, the roasted product undergoes a grinding process to reduce particle size, generating the mass of cocoa which is the precursor of other cocoa products [9].

## 4. Cocoa By-Products

Cocoa husk is the first and main residue of the cocoa industry, representing about 80% of the fruit in dry weight (d.w.). This by-product has a composition rich in lignin and non-starch polysaccharides (cellulose, hemicelluloses and pectin) (Table 1), terpenoids (chrysophanol), phenolic and carboxylic acids (protocatechuic, salicylic, citric and tartaric acids) and some free amino acids (glutamine, asparagine, serine and lysine) [15]. Serra Bonvehí and Ventura Coll (1999) evaluated fruits grown in Côte d’Ivoire, Nigeria, Cameroon, Colombia, Ecuador, Guinea and Brazil [16], verifying that geographic origin affects the protein amount (from 12.50 to 17.60 g/100 g of dry cocoa husk) [16]. The same authors also evaluated the free amino acids (315 mg/100g d.w.) lipids (3.0 g/100g d.w.), total free sugars and starch (2.80 g/100g d.w.), concluding that this by-product is a source of dietary fibre and has acceptable protein quality [16]. Vriesmann et al. (2011) analysed the composition of cocoa husks from Northeast Brazil, focusing on minerals; iron, calcium, potassium and sodium were present in mg/100 g d.w. and copper, magnesium, selenium and zinc in mg/kg [17].

The large amount of biomass generated is scattered in the soil or burned by farmers. In nature, the decomposition of cocoa husks depends on physical, chemical and biological factors [31]. Physical factors are usually temperature and sun exposure. An important chemical factor is the biomass moisture [31]. Biological factors are microorganisms, bacteria and fungi [31]. Fungi are the main biological representatives, due to their ability to decompose the complex structures of the cocoa husk as a nutritional source. Under certain climatic conditions, this biomass allows the proliferation of undesirable fungi, including potential pathogens such as *Marasmius perniciosus*, *Phytophthora palmivora* and *P. megakarya* [32]. A demonstrative example of the importance of the handling and disposing of this by-product is what happened in Brazil, in the 1980s. The fungus *Moniliophthora perniciosa*, commonly known as witch’s broom, attacked and destroyed thousands of trees, substantially reducing the production of cocoa beans for nearly twenty years [33]. Due to this fungus, there was a significant economic deficit in the region’s cocoa industry.

Cocoa pulp, also known as mucilage, is a white mass that surrounds the cocoa beans [34]. During the pre-treatment process, this white mass releases a cloudy liquid called cocoa pulp juice, produced by the action of microorganisms (mainly yeasts and acidophilic bacteria) naturally present in the fruit and in processing sites [32]. Approximately 100–150 L of cocoa pulp juice per ton of wet cocoa beans are produced during the fermentation process [32,35]. This by-product has high potential as a culture medium for microorganisms at an industrial level. Its composition is rich in sugars and minerals, without alkaloids and other toxic substances (Table 1). The production of alcoholic beverages with standardized levels of volatile compounds (higher in alcohols, esters and aldehydes) is a possible application of this product, characterized by a low concentration of methanol. Due to its aroma, this product has high acceptability and a general acceptability compared to other fruit alcoholic beverages [23,26,36,37]. Other applications in this field were tested, such as the production of soft drinks, kefir and jams. These products had a nutritional value equivalent to commercial brands and high acceptability [18,38,39,40].

Cocoa bean shell is obtained after separation of the seed, in the heat pre-treatment process or in roasting. This by-product is a lignin cellulosic complex particularly rich in dietary fibre (ranging from 18 to 60%) and other compounds (Table 1). This by-product has been studied in recent years due to its chemical composition and high potential in different industries (chemical, food, environmental) and in human health. Cocoa bean shell has adsorption capacity, high porosity and low ash content, according to studies conducted for the environmental sector. These characteristics and good mechanical performance are attractive as lignocellulosic precursors of unbound carbon monoliths [41,42]. It is also effective as a low-cost raw material for polluting adsorbents (industrial dyes, gases and heavy metals) [28,43]. In terms of human health, fractions and extracts of this by-product seem to reduce the incidence of chronic diseases (obesity, diabetes and cancer) [44,45], control eating disorders [46,47] and protect human cells against ischemic damage [24].

## 5. Nutritional Properties of Cocoa By-Products

### 5.1. Proteins and Amino Acids

The highest protein content in cocoa by-products was determined in the cocoa bean shell (16–18 g/100 g; Table 1). The levels of this nutrient in the other by-products were 0.4–6 g/100 g for cocoa pulp and 4–11 g/100 g for cocoa husk (Table 1), values that are in agreement with reports from other fruit by-products such as mango (4.28 g/100 g) and apple (5.21 g/100 g) [48]. Cocoa bean shell doubles the values mentioned, improving its potential to be valued as a source of protein [49].

In the protein characterization of the cocoa husk, 144 proteins were identified from its proteome by the Maldi-Tof/Tof-MS technique in combination with 2-DE analysis [50]. About 48% of the identified proteins are directly correlated with primary and energy metabolism. Among the identified proteins, the enzymes leucoanthocyanidin dioxygenase and anthocyanidin reductase, polyphenol oxidase and cinnamyl alcohol dehydrogenase can be highlighted, which participate in the phenylpropanoid pathway, that is, the pathway responsible for the production of secondary metabolites [50]. The presence of caffeic acid 3-*O*-methyltransferase and polyphenol oxidase can also be highlighted, which are directly correlated with lignin synthesis [50]. It is also worth mentioning that the protein content of the cocoa bean shell is very similar to that obtained in nibs. However, the vast majority of alpha amino nitrogen in the shell has a strong bond with oxidized polyphenols, producing polyphenoquinones [51,52]. The latter form covalent bonds with the NH_2_ group of proteins and, consequently, only about 1% of the protein is free [53]. Roasting has an adverse effect on protein content according to Agus et al. (1999). These authors reported a decrease (almost 10% of the total content) in the protein concentration in the cocoa bean shell after this process [54].

Reports of amino acids are scarce because there are few articles reporting such results. However, Serra Bonvehí and Ventura Coll (1999) found protein levels between 10–15% in the cocoa shell, in which 44% of the total amino acids are essential amino acids, mainly valine, leucine and lysine (Table 2) [16]. Cocoa husk, in comparison to cocoa bean shell, has half the amino acids with acid characteristics (aspartic and glutamic acid) and basic characteristics (arginine and lysine). As for proline and valine, there was an inverse variation, in which the cocoa husk has twice the concentration when compared to cocoa bean shell, while histidine, leucine and methionine have similar amounts in both.

Table 2 also shows that cocoa husk has a lower content of aromatic amino acids (phenylalanine, tyrosine, histidine and tryptophan, 0.83) than cocoa bean shell (1.26). In regard to branched chain amino acids (BCAA); essential amino acids with importance in muscle metabolism, cocoa husk is richer than cocoa bean shell (42% of essential amino acids and 16% of total amino acids comparatively to 28% and 11%, respectively).

In the cocoa bean shell, a large percentage of the amino acids are L-amino acids, and their amount increases during roasting. L-amino acids have a better nutritional value than D-amino acids [56], but these ones contribute to the flavour during Maillard’s reactions [56].

### 5.2. Lipids

Cocoa by-products showed lipid content ranging between 1.50 and 6.87 g/100 g in d.w., which is close to that reported for tomato peel fibre (6.01 g/100 g d.w.) [57].

There are very few articles on the fat extraction from the cocoa bean shell, although many authors in their articles mention this subject. Only El-Saied et al. (1981) carried out a comparative analysis of cocoa bean shell fat and cocoa butter from the same beans [58]. Table 3 and Table 4 present some physical and chemical characteristics of both, with results relatively similar to the exception for the acidity value and fatty acid composition. The higher value determined in cocoa bean shell may be due to the hydrolysis of triacylglycerols caused by the heat pre-treatment process and roasting of the beans [53] as well as from differences in the fatty acid composition (Table 4).

Fatty acid composition (Table 4) is another parameter to compare both products. A higher unsaturation of the fat of cocoa bean shell (40.6% to 34.9%) is observed, with higher values for C14:1, C16:1 and C18:2.

Through Table 4, the fatty acids that predominate in cocoa butter are oleic, stearic and palmitic acids (Figure 4), respectively, while those present in the shell are oleic, palmitic and capric acids, respectively [58]. These products have a different fatty acids profile with nutritional interest in different applications. The specific fatty acids composition of cocoa butter and their distribution on triacylglycerols are responsible for its feature in chocolate and its high value.

### 5.3. Dietary Fibres

According to the American Association of Cereal Chemists (AACC), similar to carbohydrates, dietary fibre is the edible part of plants with some resistance to digestion and absorption in the human small intestine, being complete or partially fermented in the large intestine. Dietary fibre intake has been associated with numerous beneficial health effects [60].

Table 5 shows the amount of total, insoluble and soluble dietary fibre, and the ratio between insoluble and soluble fibre in cocoa by-products. As can be seen, the lowest value for total dietary fibre was found in cocoa pulp (16.75–16.89 g/100 g of d.w.) mostly constituted by the soluble fraction (16.06–16.11 g/100 g of d.w.). Taking into account these values, cocoa pulp can be an ingredient of high interest to the food industry, due to the ability of the soluble fraction of dietary fibre to retain water, thus causing an increase in satiety after ingestion of some food, as well as helping to reduce the absorption time of some nutrients. Moreover, this by-product acts as an important thickening, gelling, foam and emulsion stabilizer agent [61].

With regard to cocoa bean shell, the total content varied between 52 and 57 g/100 g d.w., being the by-product with about 2/3 of insoluble fibre and 1/3 of soluble fibre. In the case of cocoa husk, an important fact to note is that this by-product had a ratio Insol./Sol. ranging between 13 and 18, meaning that from 55–56 g/100 g d.w. of total fibre, the majority is insoluble fibre [25]. A good balance between soluble and insoluble fractions is very important for a fibre source because the soluble fraction has a high capacity to hydrate and swell to form viscous solutions [62]. Another important fact to be highlighted is that the soluble fraction also adsorbs and retains other substances such as non-polar molecules, minerals and glucose. The insoluble fraction, on the other hand, has the ability to adsorb and retain water in its fibrous matrix, as well as to adsorb other components, however, without forming viscous solutions [63].

The total dietary fibre content of cocoa husk proved to be high: around 60% d.w., as shown in Table 6. In relation to its composition, the soluble part represents about 17% of the total content of the by-product, consisting mainly of pectins, as demonstrated by the high content of uronic acids (7.13% d.w.). Quantitatively, the insoluble part is the main component of this cocoa by-product, accounting for about 83% of the total value, in which about one third of this fraction corresponds to non-starch polysaccharides, and the remainder to lignin of Klason. Serra-Bonvehí and Aragay-Benería (1998) reported a total dietary fibre content of 43.9% in cocoa husks, with values for the soluble part higher than those analysed by Lecumberri et al. (2007) [22,64]. Regarding the constituent sugars, the insoluble fraction is rich in glucose, suggesting that cellulose was the main non-starch polysaccharide of cocoa husk fibre, about 20% of the total, which is in agreement with reports by other authors [64,65]. From the significant value obtained for uronic acids, together with the small concentration of other monosaccharides such as: arabinose, xylose, mannose and galactose, Figure 5, the possibility of the presence of hemicellulose (xyloglucans, arabinoxylans, glucuronoxylans) was noted. Quantitatively, neutral sugars and uronic acids correspond to 35.7% of the insoluble fraction. As for the soluble fraction, there is an indication of the presence of a small concentration of galactomannans, due to the appreciable amounts of galactose and mannose. These monosaccharides can still be part of pectins, which are the main component of this fraction. Regarding pectins, both in the soluble fraction and associated with insoluble polysaccharides of the cell wall in the insoluble fraction, they are determined as uronic acids and are about 20% of the total fibre of the cocoa husk. This fact is in agreement with the results reported by Redgwell et al. (2003), who noted that cellulose and pectins were the main polysaccharides in the cell wall of the cocoa husk [66].

Through Table 7 it can be seen that the values for total dietary fibre obtained by both research groups are similar, however, for total polysaccharides, a lower value was observed by Lecumberri et al. (2007) [22]. Another important fact to note about the investigation of these authors is the same reported values on the composition of dietary fibres, excluding the Klason fraction, with about 45% of pectic compounds, 35% of cellulose and 20% of hemicellulose [22], in which glucose would be the main monosaccharide that makes up this fibre. Calculations of insoluble and soluble fractions in the cocoa bean shell vary among authors; however, insoluble fractions are always the most abundant, with a ratio varying between 2.2 and 4 [25,67]. When compared to other cocoa by-products, such as cocoa husks, this by-product has equivalent amounts of total dietary fibre, but with a higher percentage of the soluble fraction, which would have more interesting biofunctional properties [25]. It can be seen that pectins are found in both fractions, soluble and insoluble, in which they are present in the form of pectins with high methoxyl contents, for the initial ones, and later in pectins with lower methoxyl content [51]. The pectin that is present in this by-product is sometimes considered a “low quality pectin”, from a comparison with commercial ones, being present in lower concentrations than in citrus fruits or apples (about 9% versus 15% and 30% d.w., respectively) [68,69].

### 5.4. Minerals and Vitamins

The ashes of cocoa husk have been studied by several research groups due to their great potential for use as a partial substitute and/or from the combination with biofuels, biogas and fertilizers [70]. Cocoa husk has a high ash content (average of 7.15 g/100 g d.w.), which can prevent oxygen from penetrating in significant amounts into the ash to reach the burning biomass, inhibiting the process [71]. The predominant mineral is potassium, which can vary between 2.5 and 7% (between the epicarp, mesocarp and endocarp). Cocoa husk from Ghana has 7% potassium, corresponding to approximately 70% of the total ash [55]. It is also worth mentioning other minerals, such as: calcium (0.3–0.8%), magnesium (0.02–0.06%), phosphorus (0.04–0.12%), sulphur (0.02–0.05%) and silicon (0.5%) [69,71]. Ashes and minerals are predominant in the epicarp, as the pericarp is more lignified [72].

Taking into account the mineral content, the cocoa bean shell has the highest concentration of minerals, as shown in Table 8. Potassium, magnesium, calcium and phosphorus are the most abundant minerals present in this material, followed by smaller amounts of sodium and iron [73]. According to the studies carried out by Bentil et al. (2012), it was observed that the fermentation, in solid state, of the cocoa bean shell, with spawning of *P. ostreatus* and *Aspergillus niger,* generated a significant increase in the concentration of calcium, phosphorus and potassium [74]. However, the mineral content in the cocoa bean shell has great variability, mainly due to its geographic origin, since the uptake of minerals by the plant is highly dependent on the availability of minerals in the soil, being, therefore, dependent on the type and the quality of the soil [75,76].

In relation to vitamins, one of the first studies reporting the cocoa bean shell as a source of vitamin D dates back to 1935, and later on, many studies were not carried out on this topic [77]. In the investigation carried out by Knapp et al. (1935), it was determined, in cocoa bean shell, a vitamin D content of about 21 IU/g (international units). It was obtained from fermented and sun-dried beans, and is currently equivalent to 0.53 µg/g, which indicates that it is 20 to 30 times higher than the value obtained in butter consumed daily [78].

Based on the results mentioned above, Kon et al. (1935) verified that cows that were fed a ration containing cocoa bean shell in its composition produced butter with higher levels of vitamin D than cows fed with a normal feed [79]. They also confirmed that vitamin D was mainly present in the fat fraction of the cocoa bean shell, containing 40% of the total vitamin D activity.

In investigations carried out by Bonvehí et al. (1998), considerable amounts of B1 and B2 vitamins were obtained in cocoa bean shell, corresponding to about 15% of the recommended dietary allowances. However, vitamins B6 and D were detected only in traces and vitamin C was not detected [64].

Concerning vitamin E, some vitamers such as α-tocopherol, (β + γ)-tocopherol and δ-tocopherol were also found at a concentration of 1.02 mg/g in the fat fraction of cocoa bean shell [80].

### 5.5. Phenolic and Antioxidant Compounds

Phenolic compounds are products of the secondary metabolism of plants, characterized as aromatic compounds that have hydroxyl groups as substituents [81].

The total phenolic compounds of the cocoa husk can vary greatly, from 2.1 to 57 mg/g in gallic acid equivalents (GAE)/g of d.w. due to several factors, such as geographic origin, variety, plant genotype, harvest time and even stress situations in the cacao tree, as well as the solvent system used in the extraction [82].

The type and time of fermentation can also affect the total phenolic compounds in the material. The fermentation generates optimal values of polyphenols after 24 h, with a reduction in these values after that time [30].

With robust equipment, such as LC-MS/MS, it is possible to identify several polyphenolic compounds in cocoa by-products. In an analysis of the cocoa husk with an extracting solution of 80% ethanol, at 40 °C and 30 min, the following phenolic compounds were quantitatively obtained: phenolic acids (protocatechuic acid and its derivatives and p-hydroxybenzoic acid), flavonoids (apigenin, ramnetine, kaempferol derivatives and flavone derivatives), luteolin and linarin [2]. The total phenolic compounds, flavonoids and flavonols of this material were, respectively, 3.24 mg GAE/g, 0.97 mg and 0.34 mg/g of epicatechin equivalents (EE)/g [83]. Among all quantified compounds catechin (36%), quercetin (21%), epicatechin (21%), gallic acid (11.3%), coumaric acid (6.5%) and protocatechin (4.5%) can be highlighted [83].

According to Karim et al. (2014), the cocoa husk extract contains an inhibition factor of some enzymes responsible for skin aging, and this fact can be correlated with the high concentration of antioxidant compounds [82]. The obtained extract presented a better performance when compared to the pine bark extract, which is normally used in cosmetic products due to its greater diversity of flavonoids and terpenoid-derived metabolites [82].

Cocoa beans are not normally consumed fresh due to their bitter taste, which is caused by the high concentration of polyphenols in the seeds [84]. From the fermentation, roasting and alkalization steps, the polyphenol content in cocoa beans can be reduced by 100 to 10% in the final product (such as chocolate). Therefore, according to Forsyth, Quesnel and Roberts (1958), the reduction and subsequent elimination of such compounds may be correlated with its diffusion out of the cotyledons [85,86].

HPLC coupled with ultraviolet (UV) detection or mass spectrometry (MS) allows detecting and quantifying a wide range of polyphenolic compounds that are specifically from cocoa by-products. Procyanidins and catechins are the main polyphenols in cocoa bean shell [87]. It can be highlighted, mainly, that (−)-epicatechin is the major compound in this material, with still appreciable concentrations of (+)-catechin, followed by its dimers, procyanidin B1 (epicatechin-(4β→8)-catechin) and procyanidin B2 (epicatechin-(4β)→8)-epicatechin), according to Table 9 and Figure 6 [88,89].

Table 10 values demonstrate the great variability of phenolic compounds in cocoa bean shell, mainly due to the polyphenolic extraction conditions and the solvents used, but the total flavonoid content and the total tannin content are generally well correlated with the values of the total phenolic compounds [89,90]. Several authors studied the various possibilities for optimizing the different types of polyphenol extraction in this material, using techniques such as supercritical CO_2_, extraction in water [91,92,93], pulsed electric fields [90], high voltage electrical discharges [94], pressurized ethanol [95] or ultrasound techniques [96]. In addition to the mentioned methods, macroporous resins have been used in order to enrich the total content of polyphenols of a cocoa bean shell extract, increasing from 2.23 to 62.87% *w*/*w* [97,98].

## 6. Applications

### 6.1. Food

When the cacao fruit is broken and opened, the beans are removed along with the pulp. Still in the field, a spontaneous fermentation process begins immediately [80]. The fermentation continues in troughs, and it is an important step to remove the mucilaginous pulp and develop chocolate flavour precursors. The pectinolytic enzymes that act at the beginning of the fermentation stage degrade pectin and, therefore, decrease the viscosity of the cocoa mass, facilitating the incorporation of oxygen and stimulating the growth of acetic bacteria that continue the fermentation process [99].

Of all the residues generated, the sweet and sour juice is drained as a result of the pulp liquefaction, due to the enzymatic action in pectin [100,101]. This translucent juice has chemical and sensory characteristics similar to the original pulp and is regionally called cocoa honey [100], also referred to in the literature as cocoa sweating and exudate [102,103]. Cocoa honey is rich in carbohydrates and total sugars; it has low fat profile (less than 3.54% of lipids) and ash content (approximately 0.2%) [104], considerable contents of calcium, magnesium and phosphorus [18] and it is acidic. The total content of dietary fibre is less than 1%, although Vásquez et al. (2019) reported 16.89% [102]. The total soluble solids content of 14.03 Brix was described for a cocoa honey sample of Brazilian origin [104]. Balladares et al. (2016) reported a higher content of total soluble solids (19.6 Brix) for an Ecuadorian cocoa honey [103]. Regarding pH, the literature reports mean values of 2.76 (Brazilian sample) and 3.58 (Ecuadorian sample), corroborating the acid taste of this product [103,104]. Natural acidity is an important factor to limit the development of spoilage microorganisms, making the medium restricted to lactic and acetic bacteria, moulds and yeasts.

As a sweet and sour juice with a natural flavour, it has reducing sugars ranging between 8 and 10% in the Brazilian sample and 6.4% in the Ecuadorian sample studied by Balladares et al. (2016) [102]. Glucose (2.13–21.4%), fructose (1.06–4.42%) and sucrose (2.13–4.06%) are the main carbohydrates found in cocoa honey [18,103,104,105]. Vitamin C levels are between 7.6 and 10.9%. Although this amount does not make the vitamin C content in cocoa honey comparable to that of fruits such as oranges and tangerines, a glass of cocoa honey juice (200 mL) corresponds to more than 20% of the recommended dietary allowances, and this amount is in the 75–110 mg range, although it varies by country, sex, and health conditions [106].

Some studies have shown that cocoa bean shell has many volatile organic compounds, with about 10 to 20% of these compounds also found in roasted cocoa beans. Due to this referred aspect, this sub-product has the potential to be used in several areas of the food industry, such as in baking with the confection of cookies and bread, as these aromatic compounds are essential for the generation of chocolate flavour and aroma [89]. Moreover, in addition to the aforementioned advantage, this material can also increase the fibre content of the food generated, providing it with some antioxidant properties [107,108]. Because of this, several investigations have been carried out in this line of research, using cocoa bean shell fat and generating a reduction in the use of vegetable oil of 50 and 70% in functional cakes and chocolate muffins, respectively, with appreciable acceptance of products by the consumer [109,110].

Due to its antioxidant properties to prevent lipid oxidation, Ismai and Yee (2006) added cocoa bean shell and roselle seed extracts to beef, proving a greater reduction in lipid oxidation, when compared to the results obtained with the synthetic antioxidant butylated hydroxytoluene and β-tocopherol as a control of the process [111].

Another study that stands out is the one carried out by Manzano et al. (2017). The authors reported an increase in the stability of soybean oil used in frying through the addition of a polyphenolic extract from the cocoa bean shell, obtaining a lower rate of peroxides and free fatty acids after a repeated use of that oil [112].

Another use for cocoa by-products was reported by Osundahunsi et al. (2007) and suggested the reintroduction of cocoa bean shell ash in the chocolate production process due to its alkalizing potential [113]. Finally, such materials were also incorporated into bioelastomers in order to preserve food for a longer period of time due to its antioxidant properties, with the creation of active packaging [114].

### 6.2. Agroindustry and Feedstuff

The use of biomass in agroindustry is increasing, being an alternative to non-renewable energy sources and as a source of valuable chemical products, requiring analyses for direct combustion, pyrolysis and anaerobic digestion, as shown in Table 11.

### 6.3. Environmental

The physical and chemical properties of cocoa by-products are potential adsorbents for the removal of undesirable compounds in industrial wastewater treatments. As shown in Table 12, one can observe some uses of these by-products, taking into account different aspects of its use.

### 6.4. Human Health

Cocoa by-products have a wide variety of uses when related to human health, in which such by-products can be used in the cosmetic field, in the fight and prevention of diseases and as antimicrobial agents, as shown in Table 13.

## 7. Conclusions

It was concluded that a number of by-products are produced during the processing of raw materials in the chocolate industry. These by-products are not necessarily a waste. If disposed incorrectly, they may generate environmental concerns, but treated correctly they become a high-value raw material for the development of new products.

Several fields of application, including human health, cosmetics, food industry and bioremediation have been suggested for the use of cocoa by-products, with very promising perspectives.

Cocoa husk is a rich source of dietary fibre and protein, as well as valuable bioactive compounds (theobromine, caffeine, flavonoids, etc.). Due to its composition, it can be used as an ingredient in food processing, or in other industries such as pharmaceuticals, cosmetics or agriculture, with a constant increase in new applications. In addition, the recovery of cocoa husks has high economic value, as it is a cheap raw material for extracting various components and can be used as biofuel.

Cocoa bean shell is inexpensive and there is an urgent need for practical and innovative ideas to exploit its full potential, increasing the overall sustainability of the cocoa agribusiness.

As changes to better efficiency and sustainability may also involve actions to add value to food-related by-products and residues, large amounts of organic compounds (e.g., pectin, antioxidants, dietary fibres and minerals) of this material justify its valorisation. They are a good source of nutrients and bioactive compounds, but their use in the food industry is minimal, in part due to limited research.

These by-products can be processed according to various functionalities and bioactivities. They are promising by-products waiting for an opportunity to improve the sustainability of the cocoa production chain.

## Figures and Tables

**Figure 1 molecules-27-01625-f001:**
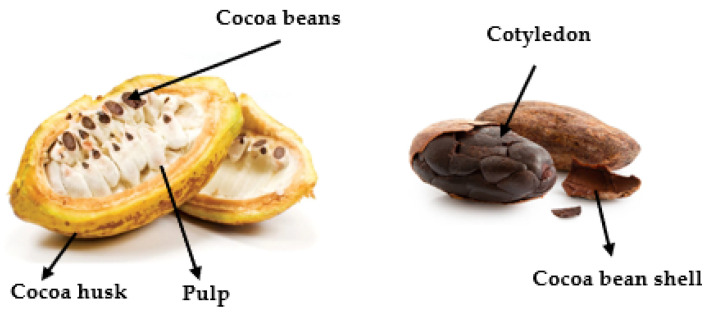
Constituents of cocoa fruit [2].

**Figure 2 molecules-27-01625-f002:**
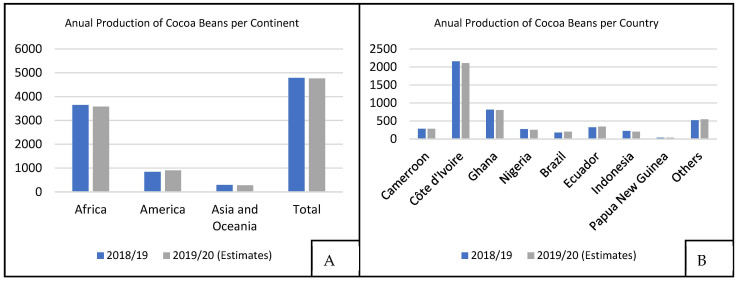
World cocoa production, at 10^3^ t, for 2018/2019 and 2019/2020 harvests, (**A**) Annual production of cocoa beans divided by each continent and (**B**) Annual production of cocoa beans divided by each country [7].

**Figure 3 molecules-27-01625-f003:**
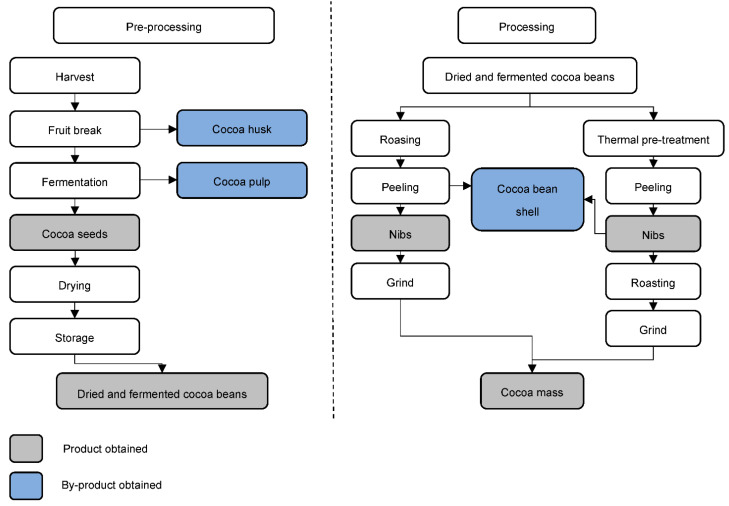
Pre-processing and processing steps for the transformation of cocoa fruit to cocoa mass [1,9].

**Figure 4 molecules-27-01625-f004:**
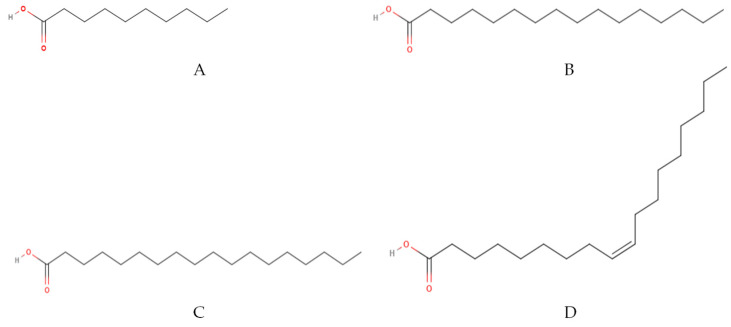
Fats most commonly found in cocoa bean shell: (**A**) Capric, (**B**) Palmitic, (**C**) Stearic and (**D**) Oleic Acid [59].

**Figure 5 molecules-27-01625-f005:**
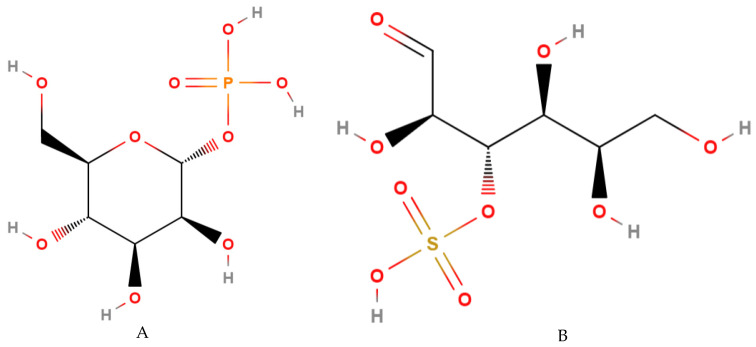
Chemical structure of (**A**) Mannose-1-phosphate and (**B**) Galactose-3-sulfate [59].

**Figure 6 molecules-27-01625-f006:**
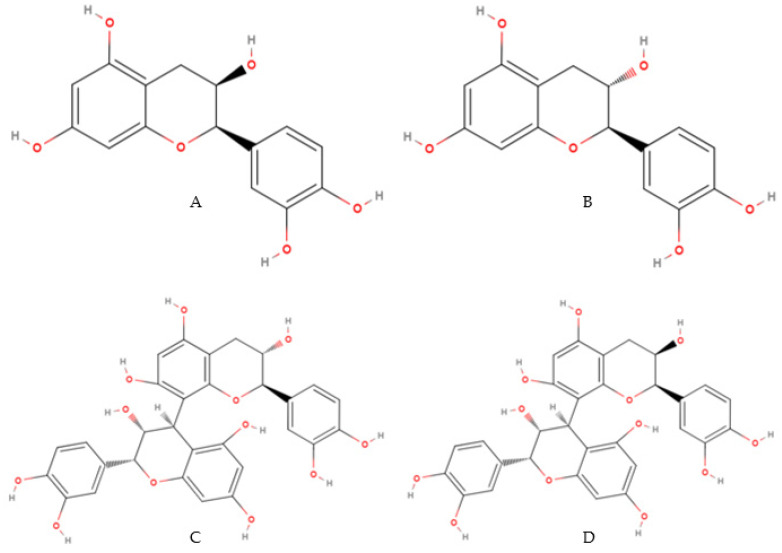
Chemical structure of catechins mostly obtained from cocoa bean shell: (**A**) Epicatechin, (**B**) Catechin, (**C**) Procyanidin B1 and (**D**) Procyanidin B2 [59].

**Table 1 molecules-27-01625-t001:** Chemical composition of cocoa by-products (g/100 g d.w.).

Compounds	Cocoa Husk	Cocoa Pulp	Cocoa Bean Shell	References
Carbohydrates	29.04–32.30	10.70–68.35	17.80–23.17	[18,19]
Cellulose	24.24–35.00	20.80–57.50	15.10	[20]
Hemicellulose	8.72–11.00	7.00–17.00	-	[21]
Lignin	14.60–26.38	12.00-14.60	32.41	[22]
Pectin	6.10–9.20	0.57–1.50	0.57–1.50	[23,24]
Total dietary fibre	36.60–56.10	16.89	18.60–60.60	[17,25]
Total proteins	4.21–10.74	0.41–5.56	15.79–18.10	[26,27]
Lipids	1.50–2.24	1.91–3.54	2.02–6.87	[18,28]
Ash	6.70–10.02	3.70–7.68	5.96–11.42	[21,27]
Minerals (mg/100 g)	3230.85	1297.07	56.75–312.57	[17]
Total organic acids	-	17.52	-	[18]
Total phenolics *	4.60–6.90	-	1.32–5.78	[29]
Anthocyanins **	-	-	0.40	[30]
Theobromine	0.34	-	1.30	[20,24]
Caffeine	-	-	0.10	[28]
Tannins	5.20	-	3.30–4.46	[22,29]
Flavonols **	-	-	1.50	[30]

* (g gallic acid equivalent/100 g) and ** (µg quercetin/100 g).

**Table 2 molecules-27-01625-t002:** Amino acids profile (g/100g) of cocoa husk and cocoa bean shell [2,16,55].

Amino Acids	Cocoa Husk	Cocoa Bean Shell
**Essential**	**2.66**	**4.15**
Arginine	0.22	0.70
Histidine	0.21	0.27
Isoleucine	0.24	0.48
Leucine	0.43	0.45
Lysine	0.40	0.79
Methionine	0.05	0.06
Phenylalanine	0.37	0.45
Threonine	0.30	0.70
Valine	0.44	0.25
**Non Essential**	**3.43**	**6.59**
Aspartic acid	0.80	1.50
Alanine	0.44	0.80
Cystine	0.09	0.25
Glycine	0.29	0.72
Glutamic acid	0.77	1.87
Proline	0.38	0.20
Serine	0.41	0.71
Tryptophan	0.04	0.12
Tyrosine	0.21	0.42
**Total amino acids**	**6.09**	**10.74**
**BCAA**	1.11	1.18
**AAA**	0.83	1.26

BCAA–Branched chain amino acids; AAA–aromatic amino acids.

**Table 3 molecules-27-01625-t003:** Fat physical and chemical characteristics of cocoa butter and cocoa bean shell [53,59].

Characteristics	Cocoa Butter	Cocoa Bean Shell
Specific gravity at 40 °C	0.9012	0.9034
Melting point (°C)	34.10	31.00
Acid value (expressed as oleic acid %)	1.68	9.12
Saponification index	191.214	205.708
Iodine index	35.57	38.73

**Table 4 molecules-27-01625-t004:** Fatty acid composition of cocoa butter and cocoa bean shell (%) [53,58].

Fatty Acid		Cocoa Butter	Cocoa Bean Shell
Capric	(C 10:0)	12.95	16.89
Lauric	(C 12:0)	Traces	Traces
Tridecanoic	(C 13:0)	Traces	Traces
Myristic	(C 14:0)	4.32	3.19
Myristoleic	(C 14:1)	1.29	2.43
Palmitic	(C 16:0)	23.31	22.27
Palmitoleic	(C 16:1)	0.95	2.55
Margaric	(C 17:0)	Traces	Traces
Stearic	(C 18:0)	24.51	12.05
Oleic	(C 18:1)	28.74	28.16
Linoleic	(C 18:2)	3.93	7.49

**Table 5 molecules-27-01625-t005:** Total, insoluble and soluble dietary fibre contents of different cocoa by-products [25].

By-Product	Total (g/100 g d.w.)	Soluble (g/100 g d.w.)	Insoluble (g/100 g d.w.)	Ratio Insol./Sol.
Cocoa pulp	16.75–16.89	16.06–16.11	0.69–0.78	0.04–0.05
Cocoa husk	55.09–56.10	2.88–4.12	51.98–53.11	12.89–18.05
Cocoa bean shell	51.88–56.70	14.53–16.24	35.64–42.17	2.45–2.53

**Table 6 molecules-27-01625-t006:** Content and composition of soluble and insoluble dietary fibre fractions from cocoa husk (% d.w.) [22].

Dietary Fibre Composition	(% Dry Weight)	
	Soluble	Insoluble
Neutral sugars ^a^	2.96	14.53
Rhamnose	0.29	0.15
Fucose	Not detected	0.06
Arabinose	0.29	0.94
Xilose	0.09	0.97
Mannose	0.51	0.96
Galactose	1.36	0.91
Glucose	0.41	10.53
*Uronic Acids*	7.13	3.48
*Klason Lignin*	-	32.41
**Total**	10.09	50.42

^a^ Sum of sugars constituents.

**Table 7 molecules-27-01625-t007:** Gravimetric determination of dietary fibre (including Klason fraction) in cocoa bean shell.

Compound	Percentage (in Dry Matter)	
	Redgwell et al. (2003) [65]	Lecumberri et al. (2007) [22]
Total dietary fibre	63.6	60.5
Total polysaccharides	38.2	28.1

**Table 8 molecules-27-01625-t008:** Mineral composition of cocoa by-products.

Minerals	Mineral Content (mg/100 g)			References
	Cocoa Husk	Cocoa Pulp	Cocoa Bean Shell	
Ca	254.00	171.50	230.00–440.00	[18,65]
Cu	6.18	-	2.35–6.62	[17,65]
Fe	5.80	-	27.60–80.50	[18,64]
K	2768.00	950.00	1250.00–1820.00	[18,65]
Mg	100.90	82.50	480.00–1290.00	[17,64]
Mn	35.72	-	4.53	[17,65]
Na	10.60	30.50	16.00–192.20	[18,64]
P	-	62.47	580.00–1000.00	[64,65]
Se	0.01	-	0.21	[17,65]
Zn	39.74	-	2.75–19.00	[17,73]

**Table 9 molecules-27-01625-t009:** Composition of catechins in cocoa bean shell [88,89].

Compounds	Concentration (mg/g)
(−)-epicatechin	0.21–34.97
(+)-catechin	0.18–4.50
Epicatechin-(4β→8)-catechin	0.55–0.83
Epicatechin-(4β)→8)-epicatechin	0.23–1.38

**Table 10 molecules-27-01625-t010:** Composition of phenolic compounds in cocoa bean shell [80].

Compounds	Concentration
Total phenolic compounds (mg GAE/g d.m.)	6.04–94.95
Total flavonoids (mg CE/g d.m.)	1.65–40.72
Total tannin (mg CE/g d.m.)	1.70–25.30

**Table 11 molecules-27-01625-t011:** Results of application of cocoa by-products in agroindustry and feedstuff.

By-Product	Methodology	Results	References
Cocoa husk	Crushed and carbonized at 400 °C for a period of 2 h.	Generation of a higher heating value (17 MJ/kg) with high ash content.	[115]
Cocoa husk	Generation of a solid base catalyst for the transesterification of soy oil into biodiesel.	Potassium from cocoa husk can be a viable base catalyst generating high yields for biodiesel production, as well as better engine performance.	[20]
Cocoa husk	Lipase immobilization through crosslinking enzymatic aggregate technology.	The immobilized enzyme is a potential catalyst for the production of biodiesel by transesterification of *Jatropha curcas* oil.	[116]
Cocoa husk	Conversion of cocoa husk through a pyrolysis process and catalytic reactions.	Production of useful chemicals such as ketones, carboxylic acids, aldehydes, furans, heterocyclic aromatics, alkylbenzenes, phenols and benzenediols.	[21]
Cocoa husk	Use and optimization of a fermentation process with the mushroom *Pleurotus ostreatus* as a biocatalyst.	After five weeks of fermentation with 0.075% (*w*/*w*) MnCl_2_, 36% increase in crude protein and total soluble carbohydrates was generated with a 17% reduction in fibre and 88% in total tannins.	[117]
Cocoa bean shell	Adsorption and desorption of phosphate-P, ammonium-N and nitrate-N in corncob biochars.	Biochar can release essential nutrients to the soil to improve being able to release PO_4_^3−^–P and weakly exchange NH^4+^–N.	[118]
Cocoa bean shell	Addition of the shell or theobromine, in different concentrations, ranging from 1, 2, 4 and 6% to the chicken feed.	In the proportions of 4 and 6% of husk there was a significant influence on the decrease in body weight of chickens and for theobromine the weight of chickens was drastically reduced.	[119]
Cocoa bean shell	Six pigs were fed a conventional cereal-based diet, or a diet obtained by substitution of 7.5% of the conventional diet with cocoa shell for 3 weeks.	An increase in microbial populations of the *Bacteroides-Prevotella* and *Faecalibacterium prausnitzii* group and a reduction in *Lactobacilli*, however, this feeding improved the proportion between the main phyla of the intestinal ecosystem.	[120]
Cocoa bean shell	This study was collected from an experimental study of performance of rabbits fed graded levels of various treatments of shell as feed supplement.	It is concluded that untreated cocoa bean shell can be used in the inclusion of 100 g/kg in the rabbit feed, while those treated with hot water can be included up to 200 g/kg in the rabbit feed for growth performance with ideal and the highest cost-benefit ratio.	[121]
Cocoa bean shell	Assessment of increased intake of sun-dried shell, with a concentration between 0 to 30%.	Reduction in average daily feed intake and egg production, as well as in spleen, kidney and ovary weight in chickens fed 25 and 30% concentration feeds, due to increased theobromine intake.	[122]

**Table 12 molecules-27-01625-t012:** Results of application of cocoa by-products in the environment.

By-Product	Methodology	Results	References
Cocoa husk	Adsorption tests were performed under agitation with different metallic elements and cocoa husk concentrations.	Efficient in removing lead from acidic solutions, with maximum adsorption after 2 h. It was also observed that the other metals do not influence lead adsorption in the matrix.	[123]
Cocoa husk	The cocoa husk (1–2 mm size) was activated with the reactive orange dye and subsequently carbonized between 500 °C and 700 °C.	The kinetics showed that the material is an effective adsorption agent with a maximum adsorption of 111 mg/g of Remazol Brilliant Black R, for its use as a dye removing agent in textile effluents.	[124]
Cocoa husk	The cocoa husk underwent an alkaline treatment (NaOH) for the adsorption of methylene blue.	The maximum adsorption capacity of methylene blue is 263.9 mg/g, where a pseudo-second order provides the best correlation to predict the kinetic process. The adsorption of methylene blue was considered endothermic and spontaneous.	[125]
Cocoa husk	Cocoa husks were used as a precursor to the activated carbon for dye removal from textile industry effluents.	The best results obtained were from the production of activated charcoal with cocoa husk, being chemically activated with ZnCl_2_ and subsequently carbonized. Removal levels reached about 80% in a period of less than 1 h with pore sizes of 0.25–1 mm.	[126]
Cocoa bean shell	Ethanol production from cocoa bean shell using acid hydrolysis and *Saccharomyces cerevisiae*.	The pH has the most relevant effect on the yield of ethanol production, followed by the fermentation time and, finally, the yeast concentration. Cocoa bean shells and the developed methodology are excellent for an optimization of ethanol production.	[127]
Cocoa bean shell	Energy use evaluation of solid biofuels (wheat straw and rapeseed) and their mixtures with suitable additives (cocoa bean shell, lignite and coal sludge).	The results of thermal emission measurements demonstrated that all samples meet the requirements for carbon monoxide, but the average emission concentrations of nitrogen oxides exceed the limits.	[118]

**Table 13 molecules-27-01625-t013:** Results of application of cocoa by-products in the human health.

By-Product	Field	Methodology	Results	References
Cocoa husk	Cosmetics	Extraction of cocoa husk with an ethanolic solvent (80%) to study the effect of skin lightening	A sun protection effect was observed from the in vitro mushroom tyrosinase assay (in the absorption range between 200–400 nm wavelength).	[82]
Cocoa husk	Cosmetics	Resveratrol and fatty acids, such as linoleic acid, were isolated from an acetone-soluble extract of cocoa husk.	In the results obtained, it was observed that such compounds have in vitro skin lightening properties and do not cause adverse effects.	[128,129]
Cocoa husk	Cosmetics	To make African Black soap, cocoa husk ash is used, along with *Cocos nucifera* (coconut oil), *Butyrospermum parkii* (raw shea butter), among others, and water.	Soap is used in environmentally friendly cleansers and conditioners.	[130]
Cocoa husk	Cosmetics	For the determination of EC50, fibroblast cells were used. Finally, the gel was tested by 12 panel members to determine the effectiveness of cocoa husk extracts in gel form using Visioscan to reduce skin wrinkles and improve skin condition.	From the results it was observed that the extract is a potential ingredient for wrinkle reduction. In which the wrinkles of the skin reduced around between 6 to 13%, between 3 and 5 weeks, still generating an increase in the hydration of the skin, around 3% after 3 weeks of application of the gel.	[131]
Cocoa husk	Antibacterial	To generate the crude extract, the cocoa husk underwent a spontaneous aerobic fermentation process. The generated extract was fractionated by solvent partition with polar solvent extraction or by silica gel chromatography, in which they were analysed for chemical composition and bioactivity.	The extract showed efficacy against Gram-negative *Salmonella choleraesuis* (ATCC10708) (1 mg/mL MIC) and Gram-positive *Staphylococcus epidermidis* (ATCC35984) (2.5 mg/mL MIC), still showing a high inhibitory activity against *Pseudomonas aeruginosa* (ATCC15442).	[100]
Cocoa bean shell	Cardiovascular diseases	Investigation of cocoa flavonols due to their antioxidant activity in plasma, causing a decrease in platelet reactivity or their anti-inflammatory properties.	It was observed that such compounds are correlated with the prevention of some diseases, such as cardiovascular diseases, due to their properties being correlated with the reduction of the potential for the emergence of atherosclerosis or thrombosis.	[45]
Cocoa bean shell	Cardiovascular diseases	Investigation of the in vivo bioavailability of cocoa bean shell from dietary intake and its contribution to cardiovascular health.	It was observed that cocoa bean shell fibre has a considerable ability to adsorb a large amount of oil and cholesterol, thus reducing its bioavailability during the digestion process.	[27]
Cocoa bean shell	Diabetes and obesity	In vivo studies were carried out, with the help of rats, to verify changes in lipid and cholesterol rates, from the ingestion of cocoa bean shell.	Significant reductions in total cholesterol and low-density lipoprotein were observed, due to the effect resulting mainly from the soluble part of dietary fibre.	[22]
Cocoa bean shell	Diabetes	The studies were carried out in vitro simulating a diabetic condition in different cell lines from or obtained from the main target tissues for the disease, to verify the efficiency of flavanols.	Cocoa flavonols act as chemo-preventive agents, helping to prevent or treat type 2 diabetes mellitus, as they regulate insulin secretion and protect pancreatic-β cells, in which they still have insulin-like activity, helping to improve glucose transport. for some organs.	[132]

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
