# Peer review of "Cocoa By-Products: Characterization of Bioactive Compounds and Beneficial Health Effects"

_molecules, 2022, doi:10.3390/molecules27051625_

Round 1
Reviewer 1 Report
The paper submitted by Soares and Oliveira is a comperhensive review of scientific literature focused on cocoa and its by-products. The manuscript is well written and worth publishing in Molecules after addressing some minor comments.
1) the authors should elaborate section 5.5 - more info on secondary metabolites occuring in the reviewed material -table with list of compounds detected or isolated should be added; chemical structures of compounds present in cocoa should be added
2) table with reviewed bioactivities should be added to section 6.4
Author Response
Reviewer 1.
Thanks for pointing out our weaknesses and for your encouraging words about our work.
The paper submitted by Soares and Oliveira is a comperhensive review of scientific literature focused on cocoa and its by-products. The manuscript is well written and worth publishing in Molecules after addressing some minor comments.
1) the authors should elaborate section 5.5 - more info on secondary metabolites occuring in the reviewed material -table with list of compounds detected or isolated should be added; chemical structures of compounds present in cocoa should be added
Phenolic compounds, which are the products of secondary plant metabolism, for the cited material were reported along lines 385 to 394, with a greater focus on compounds in higher concentration between lines 406 to 412 and in Table 9. Figure 6 includes the chemical structures of such compounds. The Figures 4 and 5 presents the chemical structure of some refered compounds in Tables 4 and 6.
2) table with reviewed bioactivities should be added to section 6.4
As required, the text from section 6.4 was transformed in the Table 13.
Reviewer 2 Report
The manuscript proposed for review is an overview of the articles, books, and dissertations published in the period 2000-2021 on cocoa by-products. The search was made with the help of Scopus, Web of Science, ScieLO, PubMed, and ResearchGate. The authors focused on cocoa transformation leading to cocoa by-products, their nutritional properties, and their applications.
The abstract looks more like an introduction than an abstract. It must be thoroughly revised to reflect the results obtained from the work performed. The authors could provide information on the database, the time period of the cited studies, and other information obtained as a summary or as a result of this review.
Do not use the word "almond" because it confuses the reader.
Author Response
Thanks for pointing out our weaknesses/mistakes of our work. We have made corrections in accordance to your suggestions.
The manuscript proposed for review is an overview of the articles, books, and dissertations published in the period 2000-2021 on cocoa by-products. The search was made with the help of Scopus, Web of Science, ScieLO, PubMed, and ResearchGate. The authors focused on cocoa transformation leading to cocoa by-products, their nutritional properties, and their applications.
The abstract looks more like an introduction than an abstract. It must be thoroughly revised to reflect the results obtained from the work performed. The authors could provide information on the database, the time period of the cited studies, and other information obtained as a summary or as a result of this review.
We revised the abstract as suggested. We also added more information about the results present in the bibliography.
Do not use the word "almond" because it confuses the reader.
This word was modified to cocoa bean or seeds, depending of the part of the text.
Reviewer 3 Report
Soares and Oliveira present a summary of cocoa by-products highlighting the potential of such derivatives to be source of myriad of researches as well as their relevance for the sustainable development of cocoa products. This is a very relevant topic related to food technology, nutrition, and circular economy that could be of interest for readers that are involved with cocoa research and development. Overall, the manuscript is well written and could be considered for publication after minor revision.
Minor concerns
1- Lines 95-96: “The drying stage is intended not only to eliminate the water in excess (final 95 moisture close to 7%),”… Authors should add a reference after this sentence to support this information. If it is related to reference 14, please include the citation after the sentence (even though it could be found on line 98).
2- Figure 3: Is there a reference that supports both work flows presented on Figure 3? If so, please add the citation in the caption.
3- Line 125: Please, add the acronym “d.w.” after “dry weight” as follows: “…80% of the fruit in dry weight (d.w.).”
4- Lines 508-511: Regarding the bacteria presented by authors, are they standard reference microorganisms? Please, include the ATCC (or similar id) for each microorganism if the information is available on reference 99.
5- Lines 523-527: Authors should add reference(s) that support(s) the sentence.
6- Despite authors have mentioned cosmetics as a potential field of application for cocoa by-products (line 539), no examples though have been presented on section 6 (Application). It is worth mentioning examples from literature about unusual cocoa derivatives that are used as cosmetic supplies.
Author Response
Thanks for pointing out our weaknesses and for your encouraging words about our work.
Soares and Oliveira present a summary of cocoa by-products highlighting the potential of such derivatives to be source of myriad of researches as well as their relevance for the sustainable development of cocoa products. This is a very relevant topic related to food technology, nutrition, and circular economy that could be of interest for readers that are involved with cocoa research and development. Overall, the manuscript is well written and could be considered for publication after minor revision.
Minor concerns
1- Lines 95-96: “The drying stage is intended not only to eliminate the water in excess (final 95 moisture close to 7%),”… Authors should add a reference after this sentence to support this information. If it is related to reference 14, please include the citation after the sentence (even though it could be found on line 98).
As required, the reference was added after the excerpt (final moisture close to 7%), in which this sentence was based on reference 14.
2- Figure 3: Is there a reference that supports both work flows presented on Figure 3? If so, please add the citation in the caption.
Figure 3 was developed from references 1 and 9, so these references were added in the figure legend.
3- Line 125: Please, add the acronym “d.w.” after “dry weight” as follows: “…80% of the fruit in dry weight (d.w.).”
As required the acronym "d.w." was added after the term “dry weight” in the phrase “…80% of the fruit by dry weight”.
4- Lines 508-511: Regarding the bacteria presented by authors, are they standard reference microorganisms? Please, include the ATCC (or similar id) for each microorganism if the information is available on reference 99.
As requested, the ATCC of each microorganism mentioned between lines 508 and 511 was added, based on the information provided in reference 99.
5- Lines 523-527: Authors should add reference(s) that support(s) the sentence.
Reference 132 was added to the sentence, between lines 523-527. This sentence in this revised version was included in table 13.
6- Despite authors have mentioned cosmetics as a potential field of application for cocoa by-products (line 539), no examples though have been presented on section 6 (Application). It is worth mentioning examples from literature about unusual cocoa derivatives that are used as cosmetic supplies.
To answer this comment, Table 13 was created, It has a column that indicates in which field of human health the waste was used. Thus, in the first 4 lines of this table, examples of the use of these by-products as inputs for cosmetics were added.